# Transmitted HIV-1 is more virulent in heterosexual individuals than men-who-have-sex-with-men

Ananthu James[1], Narendra M. Dixit[1,2]*

**1** Department of Chemical Engineering, Indian Institute of Science, Bengaluru, India, **2** Centre for Biosystems Science and Engineering, Indian Institute of Science, Bengaluru, India

* narendra@iisc.ac.in

## Abstract

Transmission bottlenecks introduce selection pressures on HIV-1 that vary with the mode of transmission. Recent studies on small cohorts have suggested that stronger selection pressures lead to fitter transmitted/founder (T/F) strains. Manifestations of this selection bias at the population level have remained elusive. Here, we analysed early CD4 cell count measurements reported from ∼340,000 infected heterosexual individuals (HET) and men-who-have-sex-with-men (MSM), across geographies, ethnicities and calendar years. The reduction in CD4 counts early in infection is reflective of the virulence of T/F strains. MSM and HET use predominant modes of transmission, namely, anal and penile-vaginal, with among the largest differences in the selection pressures at transmission across modes. Further, in most geographies, the groups show little inter-mixing, allowing for the differential selection bias to be sustained and amplified. We found that the early reduction in CD4 counts was consistently greater in HET than MSM (P<0.05). To account for inherent variations in baseline CD4 counts, we constructed a metric to quantify the extent of progression to AIDS as the ratio of the reduction in measured CD4 counts from baseline and the reduction associated with AIDS. We found that this progression corresponding to the early CD4 measurements was ∼68% for MSM and ∼87% for HET on average (P<$10^{-4}$; Cohen's d, $d_s$ = 0.36), reflecting the more severe disease caused by T/F strains in HET than MSM at the population level. Interestingly, the set-point viral load was not different between the groups ($d_s$<0.12), suggesting that MSM were more tolerant and not more resistant to their T/F strains than HET. This difference remained when we controlled for confounding factors using multivariable regression. We concluded that the different selection pressures at transmission have resulted in more virulent T/F strains in HET than MSM. These findings have implications for our understanding of HIV-1 pathogenesis, evolution, and epidemiology.

## Author summary

HIV-1 encounters a key bottleneck at the time of its transmission from one individual to another. This transmission bottleneck can differ between modes of transmission. The

---

**Data Availability Statement:** All relevant data are within the manuscript and its Supporting information files.

**Funding:** This work was supported by the DBT/ Wellcome Trust India Alliance Senior Fellowship IA/

S/14/1/501307 (NMD). URL: https://www.
indiaalliance.org/. The funders had no role in study
design, data collection and analysis, decision to
publish, or preparation of the manuscript.

**Competing interests:** The authors have declared
that no competing interests exist.

stronger this bottleneck is, the more fit the virus has to be to be successfully transmitted.
Accordingly, the transmitted/founder (T/F) strains of HIV-1 may have different fitness in
risk groups that use different modes of transmission. While studies on small cohorts do
support this notion, observations of the manifestations of this differential selection bias at
the population level have been lacking. Here, we examined reported early CD4 count mea-
surements from $\sim$ 340,000 HET and MSM, across geographies, ethnicities, and calendar
years. Early CD4 counts are a measure of the severity of the infection due to T/F strains.
HET and MSM transmit predominantly via penile-vaginal and anal modes, respectively,
and do not inter-mix significantly. Remarkably, we found that HET consistently had
lower early CD4 counts than MSM. This difference could not be attributed to potential
confounding factors, such as set-point viral load. The difference thus provided evidence
that T/F strains had evolved to be more virulent in HET than MSM at the population
level. Intervention strategies may benefit from accounting for this difference between risk
groups.

## Introduction

The bottlenecks in HIV-1 transmission result in a 'selection bias' favoring fitter transmitted/
founder (T/F) viruses over less fit ones [1, 2]. Several recent studies have presented evidence of
genetic, phenotypic, and clinical manifestations of the selection bias in small cohorts [1, 3–6].
The evidence is based on different attributes of fitness, each contributing to the establishment
of infection and progressive disease. For instance, from 137 heterosexual (HET) donor-recipi-
ent pairs, T/F viruses were found to carry higher than average frequencies of amino acids asso-
ciated with high *in vivo* fitness, in terms of protein stability, immune escape and compensation
[1]. Similarly, from 127 discordant couples, higher viral replication capacity (vRC) early in
infection was associated with faster decline of CD4 T cell counts [4].

The selection bias varies with the mode of transmission [3]. The stronger the bottlenecks,
the fitter the corresponding T/F viruses are likely to be [1, 2]. Anal intercourse is over 10-fold
more permissive on average than penile-vaginal intercourse [7]. Analysis of T/F genomes from
131 subjects revealed that the T/F genomes were under greater positive selection in heterosex-
ual individuals (HET), in whom the penile-vaginal mode predominates [8], than homosexual
men, or men-who-have-sex-with-men (MSM), who transmit predominantly through anal
intercourse [3]. Among HET, men had T/F viruses with higher predicted fitness *in vivo* than
women [1], consistent with the asymmetry of the bottlenecks between insertive and receptive
penile-vaginal intercourse [7].

An important question that follows is whether the differential selection bias across modes
of transmission is manifested at the wider population level, extending beyond the restricted
cohorts examined in the trials above. Such differential bias could contribute to variations in
disease progression and treatment outcomes and underlie the diverse trajectories of the HIV-1
pandemic across infected groups in which different modes of transmission predominate. To
answer this question, a marker of the manifestation of the fitness of the T/F viruses that is read-
ily measured across large populations is necessary. Furthermore, infected groups must be iden-
tified in which the predominant modes of transmission have substantial differences in the
associated bottlenecks, so that the implications of the selection bias are detectable with statisti-
cal significance. Here, we identified CD4 T cell counts measured early in infection as a suitable
marker meeting the above criteria and MSM and HET as the relevant risk groups. We collated
early CD4 count measurements in these groups across large populations and in different

geographies and calendar years and analyzed them to deduce the impact of the differential selection bias across modes of transmission at the population level.

## Results

### A marker and risk groups for assessing population-level transmission bias

Immediately following infection, CD4 T cell counts fall steeply, recover partially, and then settle within a few weeks/months to a value smaller than in the pre-infection state [9] (Fig 1A). Subsequent changes in the CD4 counts occur slowly, over many months to years. Thus, CD4 count measurements made early in infection tend to be close to the value to which the counts settle after the initial dynamics. These early CD4 counts are expected to be minimally affected by host-specific adaptive mutations [1] and, therefore, reflective of the effects of the T/F strains. The CD4 count is a key indicator of the severity of disease: the lower the CD4 count, the more severe the disease [9]. A pathogen that causes more severe disease is said to be more virulent [10, 11]. Fitter T/F strains tend to be more virulent; in the above data from 127 acutely infected individuals, high vRC of the T/F viruses, corresponding to high viral fitness, was associated with low CD4 counts at 3 months post-infection (which roughly coincides with the time of seroconversion) [4]. We reasoned, therefore, that fitter T/F strains would lead to lower early CD4 counts.

HET and MSM are the two major risk groups driving the global HIV-1 epidemic [9]. They use predominant modes of transmission with a substantial difference in the associated selection bias [7]. Importantly, they display little inter-mixing in most geographical regions. We inferred the latter from the distinct prevalence of HIV-1 subtypes in the two groups, which we found across calendar years and geographical regions (Fig 1B and 1C and S1 and S2 Tables): MSM in western nations are dominated by HIV-1 subtype B, whereas HET comprise a mixture of subtypes [12], with subtypes B and C the predominant ones [13]. For instance, in the United Kingdom, from 2002–2010, MSM had nearly 90% subtype B infections, whereas HET had a little over 10% subtype B. Mixing between the two groups would have led to a more similar distribution of subtypes in the two groups. The two groups thus appear to have remained largely segregated. The difference in subtype prevalences holds also in Canada, Spain, France, and other nations [14–20] (Fig 1B and S1 Table). In China, the dominant subtype is CRF01-AE, which is present at a frequency of >50% in MSM but at <40% in HET (Fig 1C and S2 Table) [21], perhaps indicative of more mixing than in Europe. (In Korea, the extent of mixing could not be ascertained using subtypes because over 80% of all infections were subtype B [22]. We therefore did not include data from Korea [23] in our analysis.) In USA, though subtype B dominates both MSM and HET [24, 25], mixing between the groups has been argued not to be common [26]. Overall, thus, little mixing between MSM and HET is evident in most geographical settings.

Together, these characteristics allow for the difference in the selection bias between the two groups to be sustained long-term, potentially amplified, and manifested in sample sizes large enough for detection with statistical significance. We thus hypothesized that the stronger selection bias associated with penile-vaginal transmission than anal transmission would result in fitter, more virulent T/F strains and, hence, lower early CD4 counts in HET than in MSM.

### CD4 counts early in infection in HET and MSM

To test this hypothesis, we collated available data of CD4 count measurements either at seroconversion or at diagnosis from all large studies [19, 27–31] ($n \gtrsim 1{,}000$), which amounted to a total of $\sim 340{,}000$ patients across four geographical regions, viz., China, Europe, the UK, and the US, followed over a total period of nearly four decades, and examined the differences

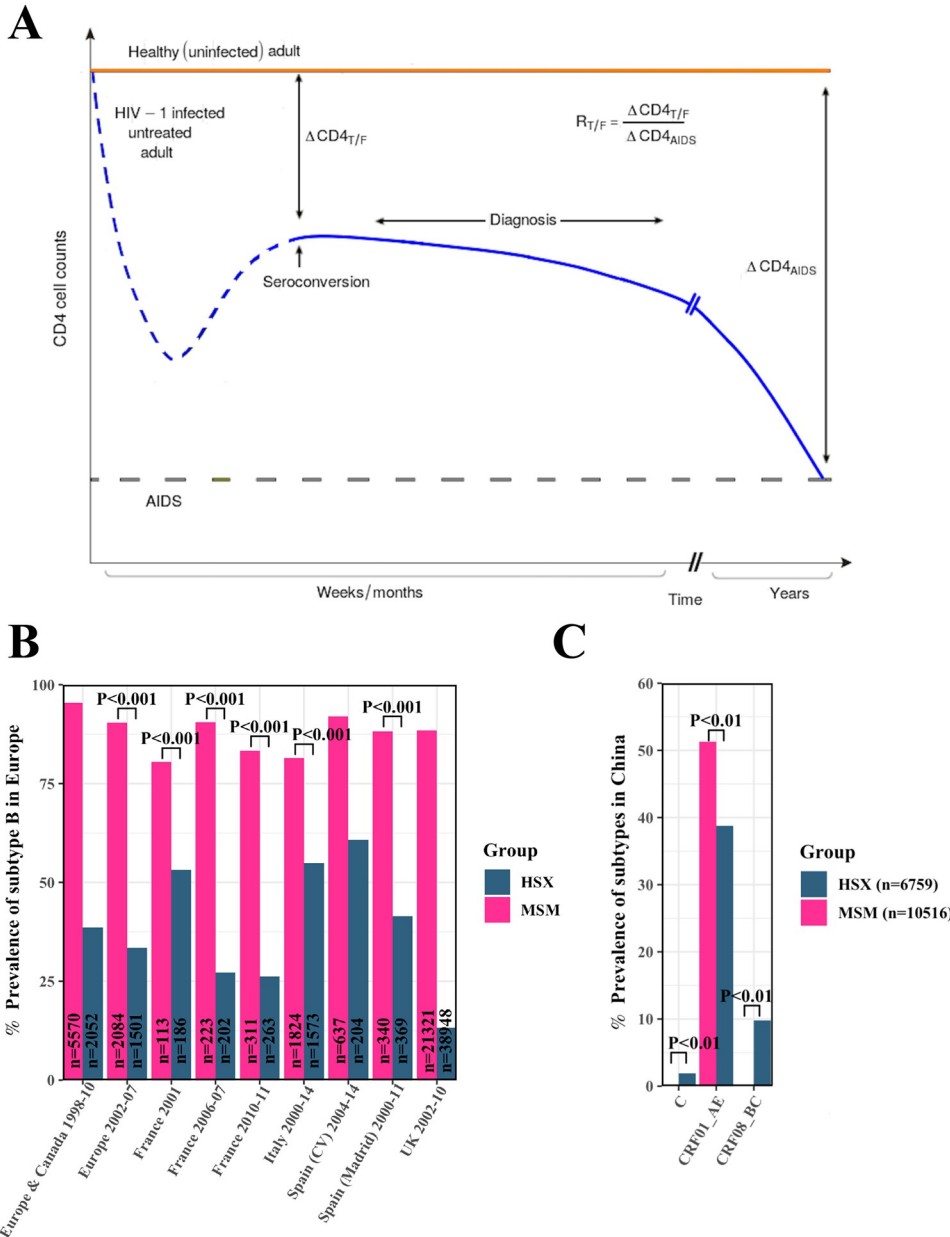

**Fig 1. Marker and risk groups for assessing selection bias at transmission. (A)** Schematic of typical CD4 count changes post HIV-1 infection (blue), before (dashed) and after (solid) diagnosis/seroconversion. The reduction at diagnosis/seroconversion relative to uninfected individuals (orange) and that associated with AIDS (grey dashed line) yields $R_{T/F}$, the reduction attributable to the T/F virus. **(B-C)** HIV-1 subtype prevalence in MSM and HET populations. Prevalence of **(B)** subtype B in different regions in Europe and Canada and **(C)** all the subtypes in China. (CV denotes Communidad Valenciana.) The sample sizes ($n$) along with the time periods of the surveys are indicated. *P* values are listed where available from the original sources (see S1 and S2 Tables).

between HET and MSM (Methods; Table 1 and S3–S5 Tables). Because sample sizes were large, we employed measures of centrality (such as mean and median) and supplemented significance estimates (P values) with effect size measures (Cohen's d, denoted $d_s$) for our analysis (Methods). Individual-patient data was not reported in these studies and was not necessary for our inferences. We found that HET consistently had lower CD4 counts than MSM (Fig 2 and

**Table 1. Early CD4 cell counts and $R_{T/F}$ in infected adults at diagnosis or seroconversion.** *P* values and Cohen's d values ($d_s$) for comparisons of the CD4 counts and $R_{T/F}$ between MSM and HET are also shown. The last row represents the population-weighted average of all the datasets. (See text and S3–S6 Tables for details and data sources).

| Sample & duration | Risk group | CD4 counts (*cells/μL*) | | | Sample size (*n*) | $R_{T/F}$ (%) | | |
|---|---|---|---|---|---|---|---|---|
| | | Mean (95% CI) | *P* value | $d_s$ | | Mean (95% CI) | *P* value | $d_s$ |
| EU/EEA 2010 – 18 | MSM* | 437 (435 – 439) | 0.00[††] | 0.54 | 60, 659 | 66.2 (65.9 – 66.5) | 0.00[††] | 0.54 |
| | HET | 302 (300 – 304)[d] | | | 55, 398 | 86.2** (85.9 – 86.5) | | |
| | HET men* | 270 (267 – 273)[d] | $5.6 \times 10^{-192}$ | 0.25 | 27, 822 | 90.0 (89.6 – 90.4) | $5.5 \times 10^{-123}$ | 0.20 |
| | HET women | 335 (332 – 338)[d] | | | 27, 576 | 82.9 (82.5 – 83.3) | | |
| USA 2006 – 15 | MSM (6 m.o. diag.)[‡‡] | 390 (386 – 394) | $3.0 \times 10^{-97}$ | 0.30 | 17, 779 | 73.7 (73.2 – 74.2) | $5.2 \times 10^{-143}$ | 0.35 |
| | HET (6 m.o. diag.)[‡‡] | 314 (308 – 320) | | | 8, 310 | 85.8 (85.0 – 86.6) | | |
| | MSM (SCs, 13–29y) | 553 (545 – 561) | $1.2 \times 10^{-18}$ | 0.21 | 6, 328 | 51.0 (49.9 – 52.1) | $1.8 \times 10^{-37}$ | 0.29 |
| | HET (SCs, 13–29y) | 483 (469 – 497) | | | 2, 958 | 64.7 (62.9 – 66.5) | | |
| China[e] 2006 – 12 | MSM | 368 (366 – 370) | 0.00[††] | 0.39 | 35, 277 | 68.0 (67.5 – 68.5) | 0.00[††] | 0.38 |
| | HET | 270 (269 – 271) | | | 143, 431 | 86.7 (86.4 – 87.0) | | |
| Europe 2002 – 07 | MSM | 426 (416 – 436) | $(< 10^{-3})^c$ | 0.57 | 2, 084 | 67.8 (66.2 – 69.4) | $2.5 \times 10^{-63}$ | 0.57 |
| | HET | 283 (270 – 296) | | | 1, 501 | 88.8 (87.0 – 90.6) | | |
| UK 1990 – 98 | MSM | 331 (325 – 337)[d] | $(< 10^{-3})^c$ | 0.41 | 6, 213 | 78.2 (77.2 – 79.2) | $3.8 \times 10^{-92}$ | 0.45 |
| | HET | 230 (220 – 240)[d] | | | 2, 637 | 96.0 (94.7 – 97.3) | | |
| Europe & Australia (CASCADE) SCs 1979 – 00 | MSM[†] (<40y) | 621 (609 – 633) | $4.3 \times 10^{-3}$ | 0.14 | 2, 570 | 40.0 (37.9 – 42.1) | 0.010 | 0.12 |
| | HET men[a] (<40y) | 576 (545 – 607) | | | 428 | 46.4 (41.4 – 51.4) | | |
| | HET women[†,a] (<40y) | 623 (599 – 647) | | | 349 | 46.4 (42.3 – 50.5) | | |
| | MSM[††] (>40y) | 578 (555 – 601) | 0.017 | 0.21 | 371 | 46.2 (42.1 – 50.3) | 0.073 | 0.16 |
| | HET men[b] (>40y) | 534 (500 – 568) | | | 62 | 52.4 (45.0 – 59.8) | | |
| | HET women[‡,b] (>40y) | 580 (548 – 612) | | | 50 | 51.8 (44.1 – 59.5) | | |
| | MSM (total) | 616 (604 – 628)[d] | 0.027 | 0.07 | 2, 941 | 40.7 (38.8 – 42.6) | $6.9 \times 10^{-4}$ | 0.12 |
| | HET (total) | 592 (570 – 614)[d] | | | 889 | 47.1 (43.7 – 50.5) | | |
| Overall | MSM | – | | | 124, 953 | 67.8 (67.5 – 68.1)[d] | 0.00[††] | 0.36 |
| | HET | – | | | 212, 166 | 86.5 (86.3 – 86.7)[d] | | |

*For the comparison between MSM and HET men, *P* = 0.00[††] for both the cell counts and $R_{T/F}$, whereas $d_s$ = 0.68 for CD4 counts and 0.63 for $R_{T/F}$.

** $R_{T/F}$ in HET changed marginally to 84.9% (95% CI = 84.6–85.2%) when we used CD4 counts in 35% of healthy individuals of 746 cells/μL from Tanzania [32] (representative of sub-Saharan Africa), instead of 941 cells/μL from Italy (representative of EU/EEA), and was still substantially higher than $R_{T/F}$ in MSM (*P* = 0.00[††]; $d_s$ = 0.49).

[††]These P values were below the lower representation limit (= $2.23 \times 10^{-308}$) of R (and Excel).

[†]*P* = 0.44 and $d_s$ = 0.01 for CD4 count and *P* = $3.1 \times 10^{-3}$ and $d_s$ = 0.12 for $R_{T/F}$ comparisons between MSM and HET women aged <40 years.

[‡]*P* = 0.46 and $d_s$ = 0.01 for CD4 count and *P* = 0.10 and $d_s$ = 0.14 for $R_{T/F}$ comparisons between MSM and HET women aged >40 years.

[‡‡]Within six months of diagnosis.

[a]*P* = 0.010 and $d_s$ = 0.16 for CD4 count and *P* = 0.50 and $d_s$ = 0.00 for $R_{T/F}$ comparisons between HET men and HET women aged <40 years.

[b]*P* = 0.026 and $d_s$ = 0.37 for CD4 count and *P* = 0.46 and $d_s$ = 0.02 for $R_{T/F}$ comparisons between HET men and HET women aged >40 years.

[c]Reported in the original sources.

[d]See Methods and S3–S6 Tables. For the CASCADE 1979–00 cohorts, the 95% CIs here are slightly different from those in S3 Table because we assumed a normal approximation here, required for the calculations of SD, $d_s$, and P value.

[e]We ruled out any bias in the results from China due to contaminated blood supplies, as HIV-1 testing before blood donation became mandatory in the mid-1990s [33, 34].

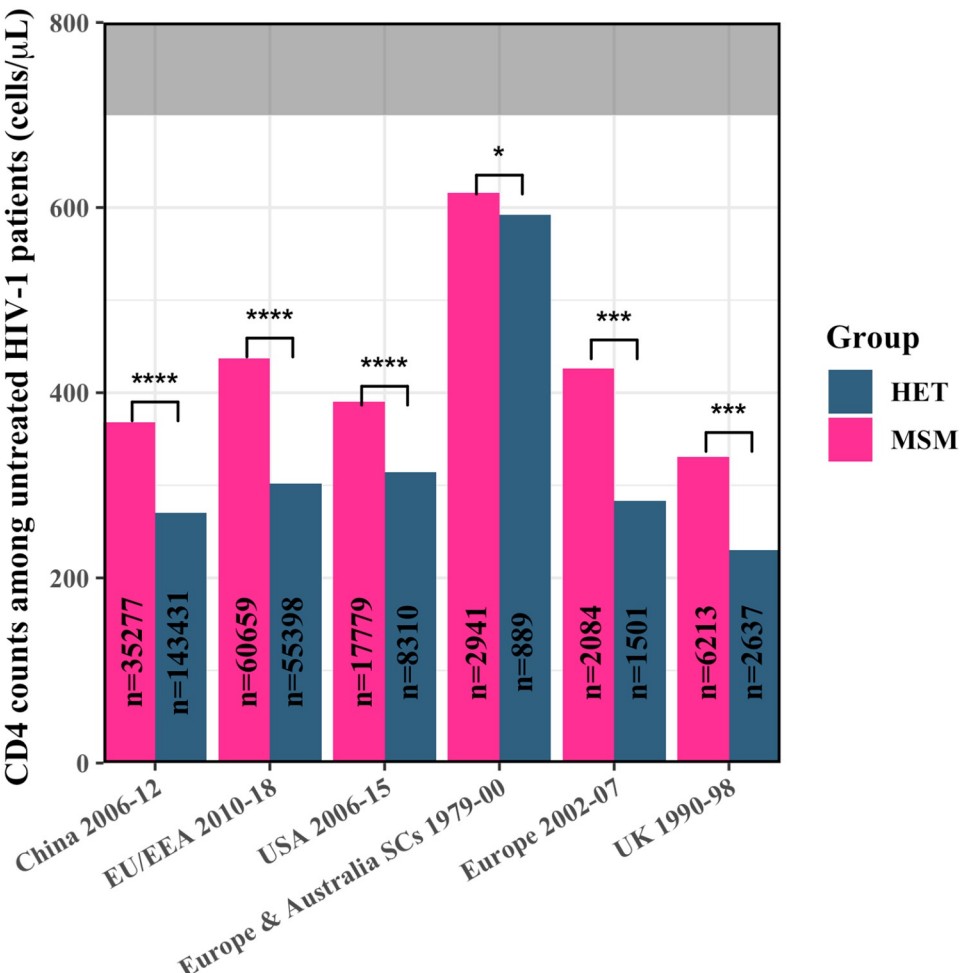

**Fig 2. Early CD4 T cell counts in MSM and HET.** Early mean CD4 cell counts in untreated infected adult HET and MSM from different geographical regions and calendar years (see Methods, Table 1 and S3–S5 Tables). The grey region indicates counts in uninfected, healthy individuals. The sample sizes ($n$) are shown. SCs are seroconverters. ****, ***, ** and * indicate $P<10^{-4}$, $P<10^{-3}$, $P<10^{-2}$ and $P<0.05$, respectively.

Table 1 and S3–S5 Tables). For instance, measurements from ∼120,000 patients across 21 countries in the European Union and European Economic Area (EU/EEA) [27] indicated, following population-weighted averaging of yearly data during 2010–2018, that the mean CD4 count in MSM at diagnosis was ∼440 cells/$\mu$L, whereas it was substantially lower, ∼300 cells/$\mu$L, in HET ($P<10^{-4}$; Cohen's $d$, $d_s$ = 0.54). The numbers were similar in the preceding 5 year period (2002–2007) reported by a smaller study involving a few thousand patients [19]. In the UK [28], measurements from close to 9,000 patients during 1990–1998 showed that the counts at diagnosis were ∼330 cells/$\mu$L in MSM and ∼230 cells/$\mu$L in HET ($P<10^{-3}$; $d_s$ = 0.41). In China [31], during 2006–2012, the mean CD4 counts at diagnosis from ∼180,000 patients were ∼370 cells/$\mu$L in MSM and ∼270 cells/$\mu$L in HET ($P<10^{-4}$; $d_s$ = 0.39). Similarly, in the US [30], from over 25,000 patients during 2006–2015, the counts at diagnosis were ∼400 cells/$\mu$L in MSM and ∼300 cells/$\mu$L in HET ($P<10^{-4}$; $d_s$ = 0.30). We also examined the counts at seroconversion where possible. Using the reported diagnosis delays and the slopes of CD4 count decline in the US population above [30], we estimated that the cell counts at seroconversion, for the age group 13–29 years, were ∼550 cells/$\mu$L in MSM and ∼480 cells/$\mu$L in HET

(P<$10^{-4}$; $d_s$ = 0.21) (Methods). In the CASCADE study [29], involving ∼4,000 patients during 1979–2000 in Europe and Australia, the mean cell counts at seroconversion were ∼620 cells/$\mu$L in MSM and ∼590 cells/$\mu$L in HET (P = 0.027; $d_s$ = 0.07). (Note that the differences become even more significant upon accounting for baseline CD4 count variations; see below.) Remarkably, although the effect sizes varied across studies, we did not find any large study that reported higher early CD4 cell counts in HET than MSM.

## Relative reduction in CD4 counts early in infection

While the evidence from absolute CD4 count comparisons was thus strong, differences in CD4 counts in healthy (uninfected) individuals across sex, ethnicity and geographical regions could render absolute CD4 counts only an approximate measure of the virulence of the T/F strains. Two individuals may have similar early CD4 counts but may still have been infected by T/F strains of different fitness if their pre-infection CD4 counts were different, with the individual with the higher pre-infection count infected by the more virulent T/F strain. To overcome this limitation, we constructed a metric to quantify the relative reduction in the CD4 cell count, $R$, corresponding to the absolute CD4 count, $T$, as $R = \frac{T_{healthy} - T}{T_{healthy} - T_{AIDS}} \times 100 = \frac{\Delta CD4}{\Delta CD4_{AIDS}}$ $\times 100$, where $T_{healthy}$ was the count pre-infection and $T_{AIDS}$ = 200 cells/$\mu$L the count defining AIDS. $R$ thus represented the reduction in CD4 counts, $\Delta CD4$, relative to the reduction signifying AIDS, $\Delta CD4_{AIDS}$. Accordingly, $R$ was 0% when $T = T_{healthy}$ and 100% when $T = T_{AIDS}$ and decreased linearly with $T$ between these extremes. $R$ was thus a more reliable indicator of disease severity than absolute CD4 counts. To use this metric, we collated measurements of $T_{healthy}$ specific to the respective geographies, ethnicities, and sexes [32, 35–39] (S6 Table). Using the latter data, we estimated $R$ corresponding to the early cell count measurements above, which we denoted as $R_{T/F}$, indicative of the relative reduction in CD4 count due to the T/F virus (Fig 3 and Table 1). The higher the $R_{T/F}$, the more virulent would be the T/F strain. (Note that $R_{T/F}$ is a static measure and is not indicative of the 'speed' of disease progression; subsequent cell count decline can be faster despite higher early CD4 counts in MSM than HET [29, 40].)

We found that in EU/EAA, during 2010–18, $R_{T/F}$ was 86.2% in HET and 66.2% in MSM (P<$10^{-4}$; $d_s$ = 0.54). During 2002–07, these numbers were 88.8% and 67.8% (P<$10^{-4}$; $d_s$ = 0.57), respectively. The corresponding numbers were 96.0% and 78.2% in the UK (P<$10^{-4}$; $d_s$ = 0.45), and 86.7% and 68.0% in China (P<$10^{-4}$; $d_s$ = 0.38). In the US, the difference was smaller but still substantial, with $R_{T/F}$ of 85.8% in HET and 73.7% in MSM (P<$10^{-4}$; $d_s$ = 0.35). At seroconversion, these numbers were 64.7% and 51.0%, respectively (P<$10^{-4}$; $d_s$ = 0.29). For the seroconverters from the CASCADE study, the trend was consistent, with $R_{T/F}$ of 47.1% in HET and 40.7% in MSM (P<$10^{-3}$; $d_s$ = 0.12).

In some studies, data was available separately for HET men and women, allowing a comparison between HET men and MSM, thus eliminating potential confounding effects of sex (Table 1). In EU/EEA, during 2010–18, $R_{T/F}$ in HET men was 90%, much higher than the 82.9% in HET women, indicating that the difference between HET and MSM was amplified upon eliminating the effect of sex. We recall that $R_{T/F}$ was 66.2% in MSM during the same period, significantly smaller than HET men (P<$10^{-4}$; $d_s$ = 0.63) and HET women (P<$10^{-4}$; $d_s$ = 0.20). Among the seroconverters in Europe and Australia, in those aged below 40 years, $R_{T/F}$ in HET men was 46.4%, higher than the 40% in MSM (P = 0.01; $d_s$ = 0.12). The difference was similar in those aged above 40 years; $R_{T/F}$ was 52.4% in HET men and 46.2% in MSM (P = 0.073; $d_s$ = 0.16). Thus, in these comparisons too, where the effects of sex, age, and diagnosis delay were eliminated, HET had a consistently higher $R_{T/F}$ than MSM.

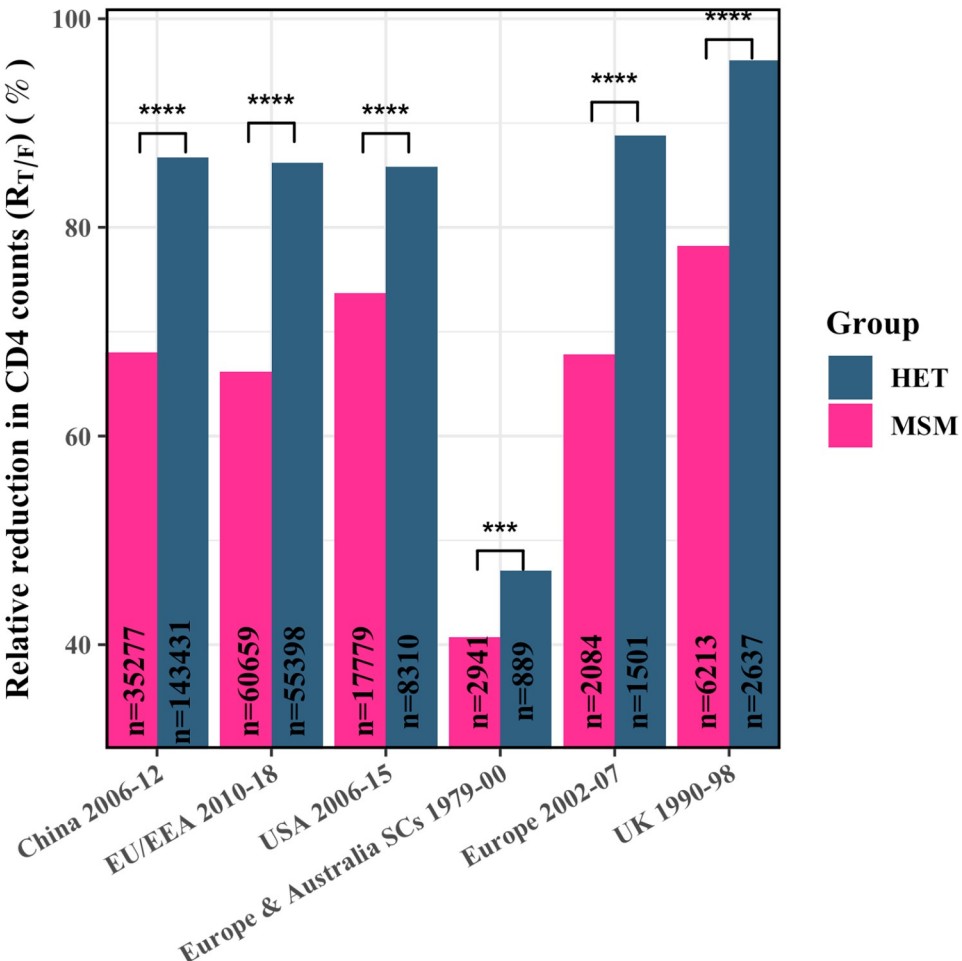

**Fig 3. The relative reduction in early CD4 counts ($R_{T/F}$) in MSM and HET.** $R_{T/F}$ in untreated infected adult HET and MSM from different geographical regions and calendar years (see Methods, Table 1 and S3–S6 Tables for details). The sample sizes (*n*) are indicated. SCs indicate seroconverters. ****, ***, ** and * indicate $P<10^{-4}$, $P<10^{-3}$, $P<10^{-2}$ and $P<0.05$, respectively.

Overall, thus, $R_{T/F}$ comparisons showed more significant differences between MSM and HET than absolute CD4 count comparisons (see Figs 2 and 3). Further, $R_{T/F}$ allowed comparison across the different datasets. Thus, while the HET all had $R_{T/F}>85\%$ at diagnosis, the MSM displayed a range from $\sim65\%$ to a little under 80%. This was comprehensive evidence of the greater virulence of the T/F virus in HET than MSM.

## Set-point viral load

To understand this finding further, we recognized that virulence can have pathogen load-dependent and -independent components [10, 11]. If pathogen load-dependent components were predominant, HET would have higher pathogen loads on average than MSM, which would then explain the higher $R_{T/F}$. The set-point viral load (SPVL) is established within weeks of infection and stays nearly constant for years [9], and is recognized as a good measure of the pathogen load in HIV-1 infection [10, 11]. A subset of the above studies reported SPVL along with early CD4 counts. We found in the latter studies that the mean SPVL was similar in MSM and HET, with the effect sizes negligible and no consistent trend towards higher SPVL in HET

(Table 2). For instance, the mean SPVL was 4.4 $\log_{10}$ copies/mL for both MSM and HET in the CASCADE study from 2006–2009 (P = 0.5; $d_s$ = 0.00). SPVL thus could not explain the differences in $R_{T/F}$ between MSM and HET. It is possible that viral loads in primary infection, before the establishment of SPVL, could be higher in HET than MSM. Measurements of viral load in primary infection are rare. A recent study did measure viral loads in primary infection in MSM and HET in a small sample ($n \sim 20$ each) and found no difference between the groups (P = 0.34) [3]. While this finding needs to be established more widely, it suggests that the differences in $R_{T/F}$ in the two groups are unlikely to have originated from differences in the pathogen load.

### Per parasite pathogenicity of T/F strains

That disease severity can be decoupled from pathogen load has been recognized previously for HIV-1 [4, 42]. A host that protects itself by suppressing the pathogen load is said to be 'resistant' to the pathogen [10]. In the context of HIV-1, greater resistance could arise, for instance, from stronger CD8 T cell responses, as suggested for elite controllers [9]. Our findings above of similar SPVL in the two groups thus suggest that MSM were not more resistant to HIV-1 than HET. A host that does not suffer disease despite high pathogen load is termed 'tolerant' to the pathogen [10]. Examples of such tolerance include HIV-1-infected viremic non-progressors, HIV-1-infected children, and SIV-infected sooty mangabeys [9, 10, 43]. MSM may thus be viewed as more tolerant to T/F strains than HET.

Hosts and pathogens can both evolve to achieve greater tolerance, as such evolution would benefit both [10]. Decoupling the contributions of the two to tolerance, however, is difficult. Nonetheless, the difference in tolerance between MSM and HET is a measure of the difference in the 'per parasite pathogenicity' of the T/F strains in the two groups. The per parasite pathogenicity measures the pathogen load-independent contribution of the pathogen to virulence [10, 11]. To quantify the per parasite pathogenicity of the T/F strains, which we denoted $P_{T/F}$, we followed the procedure in earlier studies [11] (Methods). At any given SPVL, the deviation in $R_{T/F}$ in an individual or group from that expected in the population would yield the $P_{T/F}$ in that individual or group [11]. An $R_{T/F}$ higher than expected would imply higher $P_{T/F}$. Because SPVL in the MSM and HET were close, we could directly compare $R_{T/F}$ between the two and estimate the relative $P_{T/F}$. (Accounting for the small differences in SPVL between the groups did not alter our findings; see Table 2.) Specifically, for the CASCADE data mentioned above, $R_{T/F}$ was 55.5% in HET and 44% in MSM (P $\sim 10^{-8}$; $d_s$ = 0.22 (Table 2)). The difference was a measure of the $P_{T/F}$ in HET relative to MSM. This higher $P_{T/F}$ in HET could thus be argued to have resulted in $\sim (11.5/55.5) \times 100 = 21\%$ higher virulence in HET than MSM. Similarly, for the entire duration from 2003–2009 in the same study, the higher $P_{T/F}$ corresponded to $\sim 17\%$ higher virulence in HET than MSM. In the European study from 2002–2007, the corresponding estimate was $\sim 24\%$ (Table 2). More broadly, in all the studies above (Table 1), the greater $R_{T/F}$ in HET implied, assuming similar SPVL in the two groups, a greater overall $P_{T/F}$ in HET than MSM. The greater virulence of the T/F strains was thus consistent with greater pathogenicity, reflective of the effects of the stronger selection bias at transmission, in HET than MSM.

### Confounding factors and regression analysis

Several factors could compromise our inference above of the differences in $R_{T/F}$ between HET and MSM being attributable to the differential selection bias at transmission. These factors include timing of onward transmission, diagnosis delay, HIV-1 subtype, ethnicity, sex, and age. We examined these factors next.

**Table 2. Set-point viral load and relative per parasite pathogenicity of T/F strains in HET and MSM.** SPVL*, early CD4 cell counts, the corresponding $R_{T/F}$ in infected adults at seroconversion (from the CASCADE study [41]) or diagnosis (from the Europe study [19]). For the CASCADE data, the mean and SD were calculated from the median and 95% CIs (see Methods) obtained by digitizing using WebPlotDigitizer. The 95% CIs provided here are following the normal approximation. Information for the European study is in Table 1.

| Region | Duration | Risk group | Sample size ($n$) | SPVL ($\log_{10}$ copies/mL) | | | CD4 counts (*cells/µL*) | | | $R_{T/F}$ (%) | | | $P_{T/F}$ [††,\$] |
|---|---|---|---|---|---|---|---|---|---|---|---|---|---|
| | | | | Mean (95% CI) | P value | $d_s$ | Mean (95% CI) | P value | $d_s$ | Mean (95% CI) | P value | $d_s$ | (% $R_{T/F}^{HET}$) |
| Europe & Australia SCs† (CASCADE) | 2003–05 | MSM | 2, 150 | 4.46 (4.42 − 4.50)** | $6.7 \times 10^{-3}$ | 0.09 | 584 (571 − 597)** | $3.2 \times 10^{-3}$ | 0.10 | 45.3 (43.2 − 47.4) | $4.9 \times 10^{-5}$ | 0.14 | 7.2 (13.7) |
| | | Non-MSM | 1, 141 | 4.55 (4.49 − 4.61) | | | 552 (533 − 571) | | | 52.5 (49.5 − 55.5) | | | |
| | 2006–09 | MSM | 2, 414 | 4.40 (4.36 − 4.44)** | 0.5 | 0.00 | 593 (580 − 606)** | $1.5 \times 10^{-6}$ | 0.19 | 44.0 (41.8 − 46.2) | $1.6 \times 10^{-8}$ | 0.22 | 11.5 (20.7) |
| | | Non-MSM | 818 | 4.40 (4.33 − 4.47) | | | 530 (507 − 553) | | | 55.5 (52.1 − 58.9) | | | |
| | Combined (2003 − 09) | MSM | 4, 564 | 4.43 (4.40 − 4.46)‡‡ | 0.014 | 0.06 | 589 (579 − 599)‡‡ | $1.5 \times 10^{-7}$ | 0.14 | 44.6 (43.0 − 46.2) | $3.7 \times 10^{-11}$ | 0.17 | 9.1 (16.9) |
| | | Non-MSM | 1, 959 | 4.49 (4.45 − 4.53)‡‡ | | | 543 (528 − 558)‡‡ | | | 53.7 (51.5 − 55.9) | | | |
| Europe | 2002 − 07 | MSM | 2, 084 | 4.86 (4.82 − 4.90) | $< 10^{-3}$ | 0.12 | 426 (416 − 436) | $< 10^{-3}$ | 0.57 | 67.8 (66.2 − 69.4) | $2.5 \times 10^{-63}$ | 0.57 | 21 (23.6) |
| | | HET | 1, 501 | 4.76 (4.72 − 4.80) | | | 283 (270 − 296) | | | 88.8 (87.0 − 90.6) | | | |

\* Viral load in primary infection too did not show significant differences between the groups in a study that recently reported these measurements [3]. We extracted the reported viral load data, from 16 MSM and 17 HET in Fiebig stages II and III, all infected during 1990–02 with subtype B, and found that the mean viral loads (SD) were 6.47 (0.80) and 6.38 (0.62) $\log_{10}$ copies/mL in MSM and HET, respectively (P = 0.34).

† We considered time periods in which the fractions of injection drug users and hemophiliacs in non-MSM were small [41]. Otherwise, the latter groups may have significantly different early CD4 counts from MSM and HET [29] and confound comparisons between MSM and HET. The proportions of HET in non-MSM were 78.7% and 79.8% in 2003–05 and 2006–09, respectively.

** The sample sizes for the 95% CIs to SD conversion (Methods) were set to $n/3$ and $n/4$ for the 2003–05 and 2006–09 data, respectively, assuming equal populations across years. The chosen years were proxy for the mid-value of CD4 counts in the durations.

‡‡ The largest SDs for MSM and non-MSM from the most relevant datasets.

†† This is the $P_{T/F}$ in HET relative to MSM and is estimated the difference in $R_{T/F}$ between HET (non-MSM) and MSM; see Text. The percentage contribution of $P_{T/F}$ to $R_{T/F}^{HET}$ is estimated as $\frac{P_{T/F}}{R_{T/F}^{HET}} \times 100$ and is mentioned in the brackets.

\$ We used a more detailed procedure to estimate $P_{T/F}$ by accounting for the difference in SPVL between the groups following previous studies [10, 11] (Methods). We found that the contributions varied from $\sim 10 - 25\%$, similar to the values estimated assuming no difference in the SPVL between the groups.

Early transmissions are more common to MSM than HET [44, 45]. Strong evidence of this observation comes also from the greater association of MSM with transmission clusters, which we ascertained from numerous sources [12, 18, 46–54]. A transmission cluster comprises individuals carrying viral genomes that cluster together in a phylogenetic tree [55], suggesting that the viral sequences isolated from the individuals are closely related. In Japan and China, an infected MSM had a nearly 40% chance of being part of a cluster, whereas an infected HET had <10% chance [46]. In France, the corresponding numbers were $\sim 35\%$ and $\sim 4\%$, respectively [50]. This trend was true for all the countries with data available except the Netherlands (Fig 4A). MSM also formed larger clusters than HET. The largest clusters reported in Belgium and Spain comprised nearly 100 individuals each, with the Belgian cluster containing $\sim 70$ MSM and the Spanish cluster exclusively MSM (Fig 4B) [18, 48]. Together, these data suggest greater similarity in the viral strains in MSM than HET. One way in which this greater

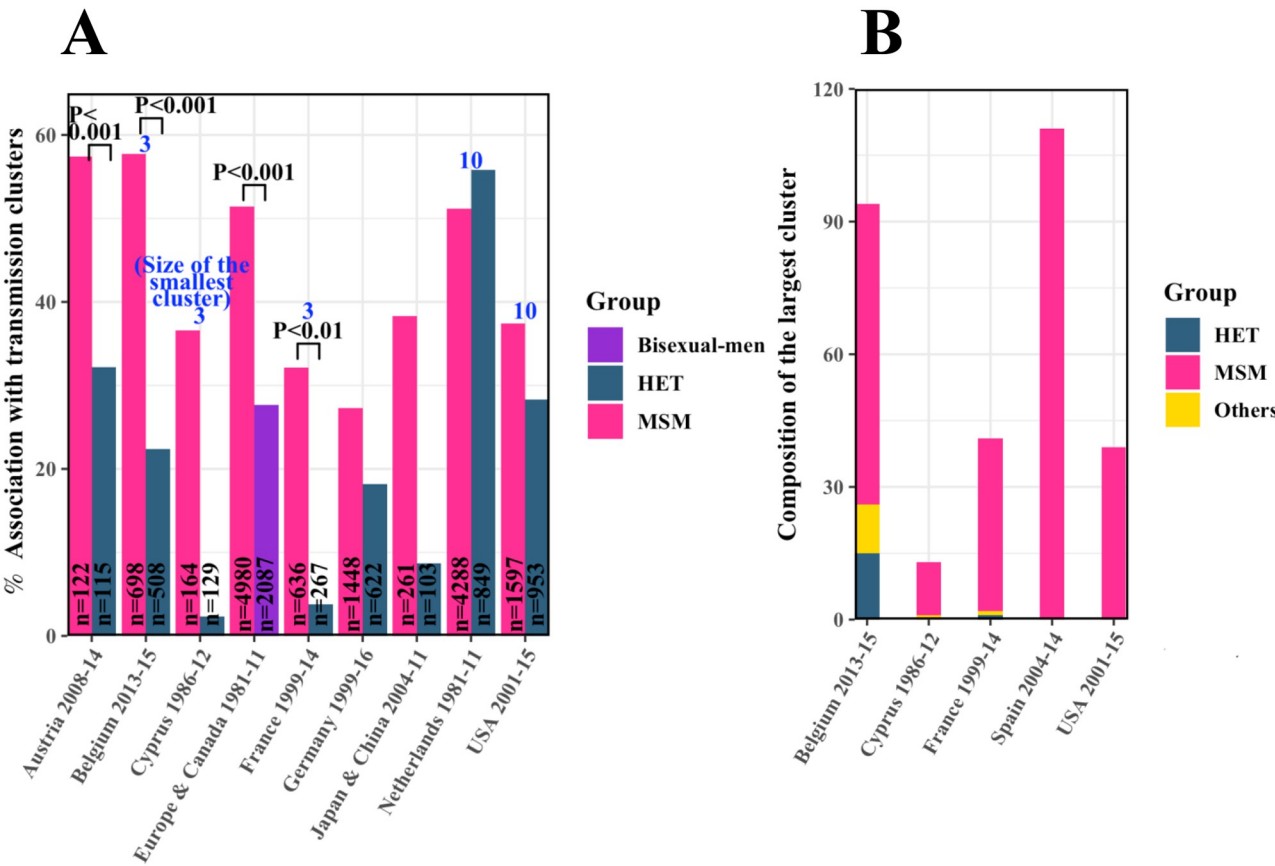

**Fig 4. Association with transmission clusters. (A)** The fraction of HET and MSM (or bisexuals in one case) associated with transmission clusters and **(B)** the size and composition of the largest clusters in different geographical regions. The sample sizes (*n*) along with the time periods of the surveys are indicated. P values and the minimum sizes of the clusters (blue text) where available from the original sources are listed (see S8 Table).

similarity could arise is by onward transmission occurring sooner after infection in MSM than HET, allowing lesser individual host-specific adaptation before transmission. This should have led to higher $R_{T/F}$ in MSM than HET, in contrast to our findings and ruling out the timing of onward transmission as a confounding factor.

Although MSM tend to be diagnosed earlier than HET [30], the differences in $R_{T/F}$ between the groups are seen also in CD4 counts at seroconversion [29, 30] (Table 1), which would occur at similar times post infection in the two groups. Besides, in China, owing to social stigma, MSM may not get diagnosed earlier than HET [31], in which case the dependence of $R_{T/F}$ on diagnosis delay in the two risk groups would be the opposite of what is expected from the corresponding selection bias. Diagnosis delay thus appeared not to be a major factor confounding our inference above. To ascertain this further, we quantified the effect of diagnosis delay on our estimates of the differences in $R_{T/F}$ between MSM and HET. In the US, the median diagnosis delays during 2006–15 were ∼ 4 years and ∼ 5.4 years for MSM and HET, respectively [30, 56]. Using the CD4 counts at diagnosis and their subsequent decline rates [30, 56], we projected the CD4 counts in MSM to 5.4 years post seroconversion (Methods), which yielded $R_{T/F}$ of 80.3%, substantially lower than that for HET of 85.8% at the same time (Table 1). Projecting the $R_{T/F}$ of HET to 4 years post seroconversion and comparing with MSM yielded similar conclusions. Thus, diagnosis delay could explain only a part of the differences in $R_{T/F}$ between MSM and HET at diagnosis. Indeed, as mentioned above, extrapolating

all the way to seroconversion, thereby eliminating effects of diagnosis delay completely, did not eliminate the differences in $R_{T/F}$ between MSM and HET (Table 1). Rather, the difference was amplified ($R_{T/F}$ was 51% in MSM and 64.7% in HET; see Table 1). (Note that with time $R_{T/F}$ in both MSM and HET would approach 100%, shrinking the difference between them and explaining the amplified difference at seroconversion.) Similarly, after extrapolating to seroconversion, substantial differences (∼20 percentage points) in $R_{T/F}$ persisted in the UK and EU/EEA cohorts as well, ruling out diagnosis delay as a major confounding factor.

Recall that MSM are predominantly infected by subtype B, whereas HET by B and C (Fig 1B). Subtype B is thought to be more virulent than subtype C [42, 44]. Moreover, in the US, where subtype B is predominant in both MSM and HET [24, 25], $R_{T/F}$ was lower in MSM (Fig 3). In agreement, subtype B T/F viruses have been found to have higher fitness in HET than MSM [3]. The effect of subtype should thus have resulted in lower CD4 counts (higher $R_{T/F}$) in MSM than HET, a trend opposite of what is observed. We quantified the effect of subtype on $R_{T/F}$ and found that a negligible portion (∼2 percentage points) of the difference in $R_{T/F}$ between MSM and HET was attributable to subtype (Methods). We concluded therefore that subtype was not a factor confounding our inference above.

In both China and the US [57], MSM and HET have similar ethnicities, and yet $R_{T/F}$ is lower in MSM than HET. In Europe (EU/EEA), while MSM are predominantly Caucasian, 30–35% of infected HET are of sub-Saharan African origin [19, 27]. Accounting for baseline CD4 count differences across ethnicities in EU/EEA did not alter our findings (Table 1), suggesting minimal effects of ethnicity on $R_{T/F}$.

The remaining factors, age and sex, could exert confounding effects. MSM are typically diagnosed at a younger age than HET. [19, 27] The younger age could result in better ability to fight disease and hence lower $R_{T/F}$ in MSM than HET. HIV-1 is known to progress differently in men and women, with women typically establishing lower SPVL but progressing faster to AIDS [58]. The latter could result in higher $R_{T/F}$ in HET than MSM.

To delineate the dependence of $R_{T/F}$ on the risk group (MSM and HET) from that on the confounding factors, we performed regression analysis (Methods). We found a small effect of age and a negligible effect of sex on $R_{T/F}$ (Table 3). These minimal effects are consistent with empirical observations. For instance, in the two European studies, MSM were 5 years [27] and 1.6 years [19] younger on average than HET at diagnosis (S5 and S7 Tables). Given the cell count decrease of ∼7 cells/μL per year of age at diagnosis [28], the early CD4 counts should have been higher in MSM by only ∼35 and ∼11 cells/μL, whereas they were higher by 135 and 143 cells/μL (Fig 2 and Table 1), respectively, a difference that could not be explained by the age at diagnosis. Similarly, healthy men had lower CD4 counts than HET and healthy women everywhere except China [32, 35–39] (S6 Table), and infected HET men displayed higher $R_{T/F}$ than MSM (Table 1), consistent with the lack of an effect of sex on the lower $R_{T/F}$

**Table 3. Results of regression analysis.** The change in $R_{T/F}$ (in %) due to variation in each factor is quantified by the associated slope. Given the dummy coding for sex (0 for female, 1 for male), $R_{T/F}$ is higher in males than females by 1.30 percentage points. Similarly, given the dummy coding for risk group (0 for MSM, 1 for HET), $R_{T/F}$ is higher in HET than MSM by ∼18 percentage points. (See Methods for details).

| Variable | Cohort | Size (n) | Slope (%) (Estimate) |
|---|---|---|---|
| Age | CASCADE 1979–00 | 3,830 | 0.24 per year |
| Sex | CASCADE 1979–00 | 889 | 1.30 |
| Risk group | China 2006–12 | 126,643 | 18.00 |
| | USA 2006–15 | 9,286 | 14.50 |
| | Combined | 135,929 | 17.80 |

in MSM. Nonetheless, the regression analysis indicated, after controlling for the effects of these latter factors, that $R_{T/F}$ was higher in HET by $\sim 18$ percentage points than MSM (Table 3). This was consistent with the difference in $R_{T/F}$ we estimated between the groups across different studies (Table 1). We concluded therefore that this difference originated from the variations in the phenotype of the T/F strains in the two groups arising from the different selection biases at transmission.

## Discussion

The bottlenecks to the transmission of HIV-1 are expected to drive HIV-1 evolution and influence the design of prevention strategies [2]. The bottlenecks are affected by the mode of transmission. Because different risk groups tend to use different predominant modes of transmission, it is possible that the T/F strains of HIV-1, directly affected by the bottlenecks, may have evolved differently in the different groups. Indeed, evidence of genetic differences between the T/F strains in MSM and HET in small cohorts has been gathered [3]. Clinical manifestations of these differences at the wider population level, however, have remained elusive. Our study shows, for the first time, that the selection bias at transmission is an important underlying factor shaping HIV-1 adaptation at the population level. The reduction in CD4 cell counts early in infection was substantially higher in HET than MSM, consistent with the more stringent selection at transmission resulting in more virulent T/F strains in HET than MSM. Our inference is based on data from several large population studies, cutting across geographies, ethnicities, and calendar years, indicating its robustness and wide applicability.

We focused on HET and MSM based on previous studies that suggested large differences in the selection bias at transmission in the two groups [7] as well as rare inter-mixing between the two groups [26]. We found strong support for the latter observation by examining the prevalence of HIV-1 subtypes in the two groups, which were vastly different in most geographies we considered. This implied that the differential adaptation of HIV-1 to MSM and HET may have been sustained and led over the years to the selection and, possibly, fixation of different adaptive mutations in the T/F viruses in the two groups, consistent with observations from small cohorts [3]. Future studies may establish them at the population level, as sequencing technologies that allow facile identification of T/F viruses emerge. The technologies may also serve to elucidate such differences between other infected groups, which are likely to be present to lower degrees than between MSM and HET, depending on the differences in the selection bias between the groups, the exclusivity of the associated modes of transmission, and the extent of mixing between the groups.

An important theory of HIV-1 evolution at the population level is the adaptation of heritable traits such as SPVL for maximizing transmission. In a series of seminal articles, an intermediate SPVL has been argued, supported by data, to maximize transmission, striking a balance between increasing transmissibility and decreasing host survival with increasing SVPL [59, 60]. If the different selection biases in HET and MSM were to lead to different dependencies of the transmissibility on SPVL, then the above theory would predict different values of the optimal SPVL in MSM and HET. From the studies we examined, a consistent trend in the differences in the SPVL values between MSM and HET did not emerge. It is possible, therefore, that the dependencies of transmissibility on SPVL may not be significantly different across MSM and HET. At the same time, previous studies have recognized that SPVL may not be the sole target of such evolutionary optimization [4, 6, 42]. For instance, HIV-1 subtype C has lower *in vivo* fitness than subtype B despite leading to higher SPVL [42, 44]. Similarly, CD4 count decline appeared independent of the SPVL during the initial years post-infection [4, 6]. The differential selection bias at transmission manifested in our study in the reduction in early

CD4 counts. Future advances to the theory of HIV-1 evolution may thus consider selection acting on multiple traits affecting transmission.

More fundamentally, whether HIV-1 is evolving in humans to be more or less virulent with time remains an unresolved question. Different studies have argued that the virulence is increasing [61], stable [62], or decreasing [44] with time. A more recent study that examined trends in CD4 counts at seroconversion in the CASCADE cohorts found that the counts declined from the early 1980s and reached a plateau around 2002, following which significant evolution was not evident [41]. The causes of these trends have remained elusive. In our present study, 4 of the 6 datasets we examined, which contained all but 13000 patients, belong to the post 2002 period (Table 1), where evolution of early CD4 counts is expected to be minimal, because of which we used counts averaged over the study periods in the respective datasets. An inference from our findings would thus be that the early CD4 counts in MSM and HET appear to have settled over the years to distinct plateau values. How the differential selection bias at transmission in the two groups, together with other potential evolutionary pressures, may have resulted in the different plateaus is an interesting question to address. Complex nested models [63, 64], which couple within-host and population-level evolution of the virus, would have to be developed, perhaps incorporating selection over multiple heritable traits, to help deduce the causes of the observed evolutionary trends.

Another aspect that warrants further investigation is the influence of mixing between groups. Although evidence of minimal mixing between MSM and HET exists, including from the comprehensive data of subtype prevalences that we have collated as part of the present study, the effect that the limited mixing may have on the differences in early CD4 counts, and hence $R_{T/F}$, between the groups remains to be quantified. Specifically, how a strain circulating in MSM, for instance, would behave when transmitted to HET, due to mixing, and vice versa, would be interesting to examine. Detailed natural history of the HIV epidemic in select countries provides some insights [65, 66]. If the population density is small, most transmission events may not lead to the establishment of an epidemic; indeed, one of approximately 25 transmission events is estimated to have resulted in a sustained epidemic in Greenland [65]. The intriguing notion of source-sink dynamics, developed first for ecological settings, has recently been proposed as relevant to natural HIV infection to assess how epidemics are sustained in geographically separated populations with diverse population densities [67]. Here, a region of high population density with a sustained epidemic can ensure the sustenance of the epidemic in a region of low population density via migration. We envision such approaches being adapted to assess the effect of mixing between risk groups. We recognize though that such models will necessarily be more involved than models aimed at describing the effects of geographical heterogeneity in population density. In the latter case, the virus and host traits are identical. With risk groups and possibly different phenotypic T/F strains, viral evolution will have to be superimposed on a spatial epidemic model.

The less stringent transmission bottleneck associated with anal transmission than penile-vaginal transmission results not only in less virulent T/F strains but may also allow a greater number of T/F viruses to establish infection in MSM than HET [68]. (Evidence that the number is not greater also exists [3].) The implications of the greater number of T/F strains remain to be elucidated. We speculate that its effect on early CD4 count decline is likely to be minimal because the latter represents a rapidly established balance between viral replication and immune control before much viral adaptation to the host. Thus, we expect our estimates of $R_{T/F}$ to remain robust to the variations in the number of T/F virions. On the other hand, the subsequent decline of CD4 counts leading to AIDS may be affected by these variations. A greater number of T/F virions may imply greater genomic diversity, which may allow easier immune escape, including via recombination [69, 70], and expedite CD4 decline. Indeed, in

some studies, MSM have been observed to exhibit faster CD4 decline despite higher initial CD4 counts than HET [29, 40].

A limitation of our study is the use of population level rather than individual level data. Currently, no dataset exists that reports measurements of all the key factors involved at the individual level in large patient cohorts. Future studies may fill this gap. Nonetheless, we recognized, based on strong independent evidence, that controlling for the effects of these factors would not substantially change the effect of risk group. Multivariable regression that controlled for confounding factors did establish the effect of risk group on early CD4 count decline. We expect our findings, therefore, to be robust.

In summary, our study presents the first large-scale evidence of a clinical manifestation of the selection bias during HIV-1 transmission, with implications for our understanding of HIV-1 pathogenesis, evolution, and epidemiology.

## Materials and methods

### Data of CD4 counts

We collated data from all large studies ($n \gtrsim 1,000$) that reported CD4 counts either at diagnosis or seroconversion in HET and MSM (Table 1 and S3–S5 Tables). From reports on countries in the EU/EEA (for the combined set of 21 countries) and China [27, 31], we digitized the median CD4 counts using WebPlotDigitizer (https://automeris.io/WebPlotDigitizer). For our analysis, we averaged the data over the study duration. To obtain sample sizes, we multiplied the diagnosed cases with the reported fraction of diagnoses, available in the annual surveillance reports [71]. The fraction was assumed to be the same across the risk groups and countries. To obtain the population-weighted average CD4 counts, we assumed that the proportions of the populations in the different transmission categories were the same across age groups and that the fractions of men and women remained conserved (except in MSM and hemophiliacs). To calculate $R_{T/F}$, we collated data of CD4 counts from healthy, uninfected adults in the USA, UK, Italy (which was used for the three studies involving European populations), Tanzania, and China (S6 Table). For $R_{T/F}$ calculations pertaining to the UK, CD4 counts from healthy MSM and HET were available, which we used. We found the counts in MSM comparable to those from healthy HET men (P = 0.22; see S6 Table). As a result, for other populations, we used the cell counts for healthy HET men when counts from healthy MSM were unavailable.

### Estimation of centrality measures

When the median, $m$, and interquartile range (IQR), $(q_l, q_u)$, of CD4 counts (or other quantities) were available, we estimated the corresponding mean, $\mu$, and standard deviation (SD), $\sigma$, using $\mu = \frac{m+q_u+q_l}{3}$ and $\sigma = \frac{q_u-q_l}{1.35}$, following a widely used method [72] applicable to large sample sizes, as considered here. When 95% confidence intervals (CIs), $(c_l, c_u)$, were available instead of IQR, we evaluated SD using another method [73] which yielded $\sigma = \frac{\sqrt{n}\,(c_u-c_l)}{3.92}$ when the sample size $n \gtrsim 100$. When CIs too were unavailable, we approximated the medians as the means, assuming the distributions to be normal. For data from China, where $\sigma$ was available for the overall population, we estimated $\sigma$ for MSM and HET using the proportion of the total $\sigma$ attributed to the two risk groups [30]. When information necessary to estimate $\sigma$ was unavailable, we used the highest $\sigma$ from related datasets. The highest $\sigma$ yielded an upper-bound on the associated P value. To estimate the SD of $R_{T/F}$, we employed the error propagation equation

[74] and derived $\sigma_{R_{T/F}} = \frac{\sqrt{\sigma_{infected}^2 (\mu_{healthy}-T_{AIDS})^2 + \sigma_{healthy}^2 (\mu_{infected}-T_{AIDS})^2}}{(\mu_{healthy}-T_{AIDS})^2}$, where $\mu$ and $\sigma$ were obtained from Table 1 and S6 Table. For $\sigma_{R_{T/F}}$ of all the data combined, we chose $\sigma$ from the CASCADE

study. Finally, for CD4 counts and $R_{T/F}$, we calculated the 95% CIs, shown in Tables 1 and 2, from the SDs.

## Estimation of CD4 counts at seroconversion

In the US study [30], a model of CD4 count decline following seroconversion has been proposed, which allowed us to estimate CD4 counts at seroconversion from measurements at diagnosis. According to the model, the CD4 count $T$ in an untreated individual at time $t$ from seroconversion follows $\sqrt{T} = a_0 + b_1 \times t + e_{1t}$, where $a_0$ and $b_1$ are constants and $e_{1t}$ is an error term. At seroconversion, the CD4 count, $T_0$, was obtained by setting $t = 0$, so that $\sqrt{T_0} = a_0 + e_{10}$. Assuming that $e_{1t} = e_{10}$, it followed that $T_0 = (\sqrt{T} - b_1 \times t)^2$. The values of $b_1$ for different age groups and transmission categories were available [56]. Also, the median delays (and IQR) in diagnosis following seroconversion, $t_d$, have been estimated [30], using which we calculated the corresponding mean and SD. For MSM and HET, we took the mid-value of the means of $t_d$ in 2006 and 2015 and chose the largest SD, and obtained $t_d = 4.05 \pm 6.67$ and $t_d = 5.40 \pm 9.04$ years, respectively, for the duration 2006–15. If $T_d$ is the CD4 count at diagnosis, then $\sqrt{T_0} = (\sqrt{T_d} - b_1 \times t_d)$. We applied the analysis to data from the most populated age group (13–29 years) and used the mid-value, 21 years, for which $b_1$ was -0.93, -0.77, and -0.80 year$^{-1}$ for MSM, HET men, and HET women, respectively. Correspondingly, we obtained $b_1 = -0.79$ year$^{-1}$ for HET. To obtain uncertainties in the estimates of $T_0$, we repeated the above analysis with $T_d$ and $t_d$ set at values $\pm\sigma$ away from their respective means, but ensuring that their lower bounds $\geq 0$ and omitting terms that are second order in $\sigma$. Half the difference between the resulting maximum and minimum values of $T_0$ yielded the $\sigma$ associated with the seroconversion data.

The above projection technique could be used to estimate CD4 counts at other times post seroconversion too. We applied it to estimate the CD4 counts in HET at the time of diagnosis in MSM and in MSM at the time of diagnosis in HET. This allowed comparisons between the two groups without the confounding effects of different diagnosis delays.

## Pairwise comparisons

To examine whether the mean CD4 counts (or $R_{T/F}$) were significantly higher (or lower) in MSM than HET, we employed the one-tailed t-test with unequal variance with the test statistic

$$t = \frac{\mu_{HET} - \mu_{MSM}}{\sqrt{\sigma_{HET}^2/n_{HET} - \sigma_{MSM}^2/n_{MSM}}} \text{ and degrees of freedom } d = \frac{\left[\frac{1}{n_{HET}} + \frac{(\sigma_{MSM}/\sigma_{HET})^2}{n_{MSM}}\right]^2}{\left[\frac{1}{n_{HET}^2(n_{HET}-1)} + \frac{(\sigma_{MSM}/\sigma_{HET})^4}{n_{MSM}^2(n_{MSM}-1)}\right]}, \text{ where } n_{HET} \text{ and}$$

$n_{MSM}$ were the two sample sizes, respectively [75]. The tests were performed using the R package [76].

## Effect size calculation

We estimated the standardised mean difference, $d_s$, known as Cohen's d [77], using $d_s = \frac{\mu_{HET} - \mu_{MSM}}{\sqrt{\frac{(n_{HET}-1)\sigma_{HET}^2 + (n_{MSM}-1)\sigma_{MSM}^2}{n_{HET} + n_{MSM} - 2}}}$. Depending on its magnitude, the effect is broadly classified as negligible ($|d_s| \leq 0.2$), small ($0.2 < |d_s| \leq 0.5$), medium ($0.5 < |d_s| \leq 0.8$), or large ($|d_s| > 0.8$). Note that the classifications are not universal and are to be interpreted along with other supporting evidence [77].

## Data of set-point viral loads

From among the studies that reported early CD4 counts in MSM and HET above, we collated corresponding data of SPVL where available (Fig 2). Data were available for MSM and the full population in 2003–05 and 2006–09 in the CASCADE cohorts [41], and for MSM and HET in the European study from 2002–2007 [19]. For the former, we obtained $\mu$ for MSM and non-MSM, with the latter containing $\sim 80\%$ HET. Using the early CD4 counts in the various groups, we estimated $R_{T/F}$.

## Estimation of per parasite pathogenicity

To estimate the relative per parasite pathogenicity, $P_{T/F}$, we adopted the procedure developed earlier [10, 11]. Accordingly, one would deduce the dependence of $R_{T/F}$ on SPVL, or tolerance, in one group. Using the dependence, the $R_{T/F}$ of the group at the SPVL in the other group would be predicted. The deviation of the predicted $R_{T/F}$ from that measured in the other group would yield the relative $P_{T/F}$. This procedure would thus quantify the effect of the T/F strain on $R_{T/F}$ beyond that expected from SPVL. As the SPVL were similar in MSM and HET, it followed that the difference in the measured $R_{T/F}$ between the groups would yield accurate estimates of $P_{T/F}$. We ascertained this by also following the above procedure explicitly. We estimated the tolerance of MSM as $\alpha = \frac{R_{T/F}^{MSM}}{(SPVL^{MSM})^2}$, assuming that CD4 decline (and hence $R_{T/F}$) was proportional to the square of SPVL [10, 11]. Assuming a linear dependence instead did not alter our conclusions. The per-parasite pathogenicity [11] of HET relative to MSM was then $P_{T/F} = \frac{R_{T/F}^{HET}}{(SPVL^{HET})^2} - \alpha$. We quantified the percentage contribution of $P_{T/F}$ to $R_{T/F}^{HET}$ as

$\frac{P_{T/F} \times (SPVL^{HET})^2}{R_{T/F}^{HET}} \times 100 = \frac{R_{T/F}^{HET} - R_{T/F}^{MSM} \times \left(\frac{SPVL^{HET}}{SPVL^{MSM}}\right)^2}{R_{T/F}^{HET}} \times 100$. We found that the contributions varied from $\sim 10 - 25\%$, which were similar to the values obtained by assuming that the SPVL were identical in the two groups (Table 2).

## Data of HIV-1 subtype prevalence

To assess the extent of mixing between MSM and HET, we collated data of the prevalence of HIV-1 subtypes in the two groups across relevant geographical regions and calendar years. The data are summarized in Fig 1B and 1C.

## Data of association with transmission clusters

Finally, we considered the extent of association of MSM and HET with transmission clusters as an indicator of the time of onward transmission post-infection. The corresponding data we collated along with data of the compositions of the largest transmission clusters are in Fig 4.

## Multivariable regression

We performed multivariable regression using the linear model [78, 79] to estimate the effects of age, sex, and risk group on the outcome ($R_{T/F}$). Accordingly, we wrote

$$R_{T/F} = \beta_0 + \beta_1 X_1 + \beta_2 X_2 + \beta_3 X_3 \tag{1}$$

where $\beta_1$, $\beta_2$, and $\beta_3$ are the effects of age, sex, and risk group, respectively, on $R_{T/F}$, and $\beta_0$ accounts for factors not considered. $X_1$ is the mean age, and $X_2$ and $X_3$ are categorical variables representing sex and risk group, respectively. We set $X_2 = 0$ for female and $X_2 = 1$ for male,

and $X_3 = 0$ for MSM and $X_3 = 1$ for HET. To estimate $\beta_1$, $\beta_2$, and $\beta_3$, we adopted the following procedure.

We first considered the CASCADE 1979–00 seroconverter (SC) cohorts [29], which have reported data by categories aged <40 y and >40 y (Table 1). To find the mean age in the two categories, $X_{1,<40}$ and $X_{1,>40}$, we employed the reported mean age $X_1$ for the overall sample of size $n$, and the reported sample sizes in the two categories, $n_{<40}$ and $n_{>40}$, and wrote $n \times X_1 = n_{<40} \times X_{1,<40} + n_{>40} \times X_{1,>40}$. Further, we let $X_{1,<40}$ and $X_{1,>40}$ be equally removed from 40 so that $X_{1,<40} + X_{1,>40} = 2 \times 40$. We solved the latter two equations and obtained $X_{1,<40}$ and $X_{1,>40}$. We applied this process separately for MSM, HET men and HET women. The data is collated in S7 Table. By subtracting Eq (1) applied to groups above and below 40 years, but within the same risk and sex groups, we obtained equations in $\beta_1$. For instance, considering MSM (so that $X_2 = 1$ and $X_3 = 0$), we obtained $R_{T/F}^{>40} - R_{T/F}^{<40} = \beta_1(X_{1,>40} - X_{1,<40})$. Using population-weighted averages (equivalent to a best-fit) across MSM, HET men and HET women in the CASCADE cohorts, we obtained the $\beta_1$. For the individual groups, namely MSM ($n = 2,941$), HET men ($n = 490$), and HET women ($n = 399$), we obtained $\beta_1$ to be 0.25%, 0.25%, and 0.16% per year, respectively, yielding the average $\beta_1 = 0.24\%$ per year.

To examine the effect of sex, we again considered the CASCADE 1979–00 seroconverter (SC) cohorts [29], but now examined patients aged <40 years among HET men and women and patients aged >40 years in the same groups. Applying Eq (1) to the <40 years groups and subtracting yielded $R_{T/F}^{men} - R_{T/F}^{women} = \beta_1(X_{1,men} - X_{1,women}) + \beta_2$. Using $\beta_1$ estimated above, the latter equation can be solved for $\beta_2$. A similar procedure was followed for groups aged >40 years. We obtained $\beta_2$ for groups aged >40 years ($n = 112$) and <40 years ($n = 777$) to be 1.82% and 1.22%, respectively, yielding a population-weighted average $\beta_2$ of 1.3%.

To estimate $\beta_3$, we first chose the Chinese cohort, where only age and sex would differ across HET and MSM. (Subtype is not a confounding factor because subtype CRF01_AE, which infects >50% of MSM and ~40% of HET in China [21], is associated with a lower CD4 cell count than subtypes CRF07_BC and B [80] and higher viral load in MSM [80].) In the Chinese population examined ($n = 218,039$), ~30.4% ($n = 66,262$) were women [31], yielding the fraction of women among HET of $f_w \sim 36.3\%$. The overall $R_{T/F}$ in HET would obey

$$R_{T/F}^{HET} = (1 - f_w) \times R_{T/F}^{HET\ men} + f_w \times R_{T/F}^{HET\ women} . \tag{2}$$

Recognizing that $R_{T/F}^{HET\ men} = \beta_2 + R_{T/F}^{HET\ women}$, using Eq (2), we obtained

$$R_{T/F}^{HET\ men} = R_{T/F}^{HET} + f_w \times \beta_2 . \tag{3}$$

Therefore, in China, $R_{T/F}^{HET\ women} = 85.9\%$ and $R_{T/F}^{HET\ men} = 87.2\%$. With these estimates, we applied Eq (1) to MSM and HET men and subtracted the resulting equations to obtain $R_{T/F}^{HET\ men} - R_{T/F}^{MSM} = \beta_1(X_{1,HET\ men} - X_{1,MSM}) + \beta_3$. The difference in $R_{T/F}$ between HET men and MSM is 19.2% (S7 Table). MSM are generally younger than HET by 2–5 years. Accounting for this age difference, we estimated $\beta_3 = 18\%$ ($n = 126,643$).

We next repeated the above exercise with the US cohort. Since ethnicities are comparable in HIV-1 infected MSM and HET in the US [57], these two groups at seroconversion differ only by sex. Using $f_w \sim 64.0\%$ [30] in Eqs (2) and (3) along with $R_{T/F}^{HET} = 64.7\%$ (S7 Table), we obtained $R_{T/F}^{HET\ men} = 65.5\%$ and $R_{T/F}^{HET\ women} = 64.0\%$ at seroconversion. Given that $R_{T/F}^{MSM} = 51\%$ (S7 Table), it followed that $\beta_3 = 14.5\%$ ($n = 9,286$), similar to the estimate from China above. Averaging the estimates from China and the US, we calculated an overall $\beta_3 = 17.8\%$, explaining the higher virulence of T/F strains in HET than MSM after controlling for confounding factors.

In the above calculations/cohorts, HIV-1 subtype was not a confounding factor. To estimate the potential contribution from subtype, we extended the regression analysis to include subtype and applied it to the EU/EEA cohort, where subtypes differ between MSM and HET. We thus wrote $R_{T/F} = \beta_0 + \beta_1 X_1 + \beta_2 X_2 + \beta_3 X_3 + \beta_4 X_4$, where $\beta_4$ is the effect of subtype on $R_{T/F}$ and $X_4$ is a categorical variable representing subtype. We set $X_4 = 0$ for subtype B, dominant in MSM, and $X_4 = 1$ for the collective of subtypes present in HET. The other terms were the same as in Eq (1). We used the diagnosis delays of 3 years for MSM and 4.9 years for HET men [81] and, following the procedure above, estimated the CD4 counts at seroconversion to be 412 cells/$\mu$L in HET men ($R_{T/F}$ = 69.8%) and 561 cells/$\mu$L in MSM ($R_{T/F}$ = 48.6%). Using estimates of $\beta_1$ and $\beta_3$ above together with the latter estimates of $R_{T/F}$, we calculated $\beta_4$ = 2.2 percentage points, indicating a negligible effect of subtype on $R_{T/F}$.

## Supporting information

**S1 Table. Prevalence of HIV-1 subtype B in Europe.** Data from different regions in Europe show substantially higher subtype B percentage prevalence in MSM than HET ($P < 0.001$ in each study, unless specified). The sample sizes ($n$) are in parantheses.
(PDF)

**S2 Table. Prevalence of HIV-1 subtypes in China.** A recent review [21] of 130 published articles, together involving of 10,516 MSM and 6,759 HET individuals, has examined the prevalence of different subtypes in China, which is reproduced below. $P$ values indicate significant differences in the prevalences of 3 subtypes.
(PDF)

**S3 Table. Early median CD4 cell counts in infected adults from several large population studies.** The sources of the studies, the periods of study, measurement times, and other details are mentioned. The USA and European studies report IQRs, whereas the CASCADE study provides 95% CIs. NA—not available.
(PDF)

**S4 Table. Early median CD4 cell counts in infected MSM and HET from China [31].** The sample sizes ($n$) were reported, while cell counts were estimated using WebPlotDigitizer (https://automeris.io/WebPlotDigitizer). The last row represents estimates (see Methods) for the entire period 2006–12.
(PDF)

**S5 Table. Early median CD4 cell counts in infected adults from EU/EEA.** Data from HIV/AIDS surveillance in Europe during 2009–2018 is used [27]. The cell counts were estimated using WebPlotDigitizer (https://automeris.io/WebPlotDigitizer). The median ages, where available, are provided. The last row provides the mean cell counts (with SDs) and total numbers of MSM, HET men and women, respectively, estimated as in Methods.
(PDF)

**S6 Table. CD4 T cell counts in healthy adults.** Mean CD4 counts in healthy adults from different population groups which define baseline counts for estimating the relative reduction in early cell count following HIV-1 infection. Sample sizes are in brackets. SD is standard deviation.
(PDF)

**S7 Table. The complete list of categorical and continuous variables.** This list includes all the populations we considered in the study. Where available, populations are divided into age-

groups in addition to their combined set, as with the CASCADE 1979–00 datasets. For the continuous variables, the mean values are given where available. The diagnosis delay is expressed in years, whereas the viral load (SPVL) in $\log_{10}$ copies/mL. A diagnosis delay of 0.25 years is used for seroconversion. Details regarding the estimation of age are given in the text. The rows titled CASCADE 2003–05, 2006–09, and 2003–09 contain the groups that were mentioned in detail in Table 2 in the context of viral load data. The last four rows are created using the information on age in S5 Table.
(PDF)

**S8 Table. Association of MSM and HET with transmission clusters.** The percentages of individuals found to be associated with transmission clusters in MSM and HET in several studies are collated. In the second column, the numbers in parantheses indicate sample sizes examined. Where available, P values and the largest cluster sizes are indicated as other details.
(PDF)

## Acknowledgments

We thank Pranesh Padmanabhan, Rajat Desikan, Pradeep Nagaraja, Roland Regoes, and Judith Bouman for comments.

## Author Contributions

**Conceptualization:** Ananthu James, Narendra M. Dixit.

**Data curation:** Ananthu James.

**Formal analysis:** Ananthu James.

**Funding acquisition:** Narendra M. Dixit.

**Methodology:** Ananthu James, Narendra M. Dixit.

**Supervision:** Narendra M. Dixit.

**Writing – original draft:** Ananthu James, Narendra M. Dixit.

**Writing – review & editing:** Ananthu James, Narendra M. Dixit.

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
