## [Decision Letter · Decision Letter 0]

1 Nov 2021

Dear Dr. Dixit,

Thank you very much for submitting your manuscript "Transmitted HIV-1 is more virulent in heterosexual individuals than men-who-have-sex-with-men" for consideration at PLOS Pathogens. As with all papers reviewed by the journal, your manuscript was reviewed by members of the editorial board and by several independent reviewers. In light of the reviews (below this email), we would like to invite the resubmission of a significantly-revised version that takes into account the reviewers' comments.

We cannot make any decision about publication until we have seen the revised manuscript and your response to the reviewers' comments. Your revised manuscript is also likely to be sent to reviewers for further evaluation.

Sincerely,

Daniel C. Douek

Associate Editor

PLOS Pathogens

Thomas Hope

Section Editor

PLOS Pathogens

Kasturi Haldar

Editor-in-Chief

PLOS Pathogens

orcid.org/0000-0001-5065-158X

Michael Malim

Editor-in-Chief

PLOS Pathogens

orcid.org/0000-0002-7699-2064

Reviewer's Responses to Questions

**Part I - Summary**

Reviewer #1: This manuscript investigates the possible effect of within risk group transmission on the HIV virulence. This is a previously looked at association, which the authors appropriately cite, and here they investigate a large collection of data. The authors find evidence that suggest that HIV spread among MSM vs HSX differ such that MSM virus cause less severe infections than HSX virus.

Reviewer #2: This is an interesting manuscript that investigates differences in CD4 T cell count at the time of HIV infection diagnosis or seroconversion, as a measure of virulence, in men who have sex with men (MSM) and heterosexuals (HSX). The authors hypothesize that such differences are the result of differential selection bias during transmission and examine early CD4 T cell count measurements reported from 340,000 infected HSX individuals and MSM, across geographies, ethnicities and calendar year. The authors’ conclusion is that, because MSM have higher CD4 counts at the time of diagnosis and seroconversion than HSX, this reflects the greater selection bias for more fit viruses during penile/vaginal sex compared to anal sex. Surprisingly, this increased “virulence” was not reflected in higher VL in HSX individuals vs MSM. This large meta-analysis is quite novel and yields results that support differences in viral pathogenicity developing over time in different risk populations.

**Part II – Major Issues: Key Experiments Required for Acceptance**

Reviewer #1: While the authors do address confounding issues, one issue stands out as more severe than the others, namely the fact that HSX are diagnosed significantly later than MSM (for sure in US and EU). The authors qualitatively and anecdotally comment on this, but do not attempt to quantitate this effect as they do with other potentially confounding factors. The later diagnosis would certainly bias the results to see HSX with lower CD counts and more advanced disease at time of diagnosis.

If the effect of within risk group transmission was an ongoing process (and there is little cross-transmission between MSM and HSX, as shown), then why is the difference in virulence not changing over time? The apparent lack of HSX-virus getting more virulent over time suggests that the difference in MSM and HSX virus is not an effect of a process but rather something that does not change, like the fact that HSX are diagnosed later (with a similar delay over time).

Even if MSM and HSX transmissions mostly happen within these risk groups there is transfer between them. Depending how frequent and along what type of network such transfer happens, different results are expected when measuring within the risk groups. Can the authors address this with some (compartmental?) modeling or at least discuss how that complicates detection of differences qualitatively, using previous research on the connections between risk groups? Could this explain why there are no time trends in virulence?

Reviewer #2: Overall, the results presented from this very large multi-cohort analysis do point to a greater impact of early HIV-1 infection on CD4 T cell counts in HSX individuals compared to MSM. Nevertheless, for most European countries (including the UK) there is a significant fraction of HIV-1 infections initiated by non-B subtype viruses in the HSX cohorts and while the authors point to data from the US, where subtype B is predominant in both groups, and a couple of papers suggesting that subtype B is more pathogenic, they do not test this formally using statistical analyses. Since for most cohorts the subtype composition is significantly different between MSM and HSX, this analysis should be performed.

**Part III – Minor Issues: Editorial and Data Presentation Modifications**

Reviewer #1: The risk of MSM infection is greater than that of HSX; and the MSM bottleneck lets >1 virus through more often than in HSX (Leitner & Romero-Severson, Nat Microbiol 2018). Thus, the genetic bottleneck is less strict in MSM. How may that affect the issue studied here?

HET is a more common abbreviation of heterosexual (transmission) than HSX.

Reviewer #2: 1. Despite the fact that the authors show in Table S6 that, with the exception of China, healthy women generally exhibit higher CD4 T cell counts than men, this is not fully developed in the paper. For the most part men and women are lumped together in the HSX category – rather than being considered separately where the data are available. A direct comparison of MSM with HSX men would seem to be a much more direct comparison and likely would reveal an even greater impact on CD4 counts and R values between the two groups. This data is presented for EU/EAA 2010-18 at diagnosis and CASCADE seroconverters in Table 1 but is not developed or discussed in detail in the paper.

2. The significance of the per parasite pathogenicity analysis is not clear. In many ways the pathogenicity values derived in this section seem to be simply restating in a somewhat redundant way the difference in R between the two groups – a better rationale for defining this term as an independent value is needed.

3. The consistency of the results from China with those for other countries is somewhat surprising given that many of the initial heterosexual cases likely resulted from contaminated blood supplies. This is not discussed in the text.

PLOS authors have the option to publish the peer review history of their article (what does this mean?). If published, this will include your full peer review and any attached files.

Reviewer #1: No

Reviewer #2: No
---

## [Decision Letter · Decision Letter 1]

27 Jan 2022

Dear Dr. Dixit,

We are pleased to inform you that your manuscript 'Transmitted HIV-1 is more virulent in heterosexual individuals than men-who-have-sex-with-men' has been provisionally accepted for publication in PLOS Pathogens.

Best regards,

Daniel C. Douek

Associate Editor

PLOS Pathogens

Thomas Hope

Section Editor

PLOS Pathogens

Kasturi Haldar

Editor-in-Chief

PLOS Pathogens

orcid.org/0000-0001-5065-158X

Michael Malim

Editor-in-Chief

PLOS Pathogens

orcid.org/0000-0002-7699-2064

Reviewer Comments (if any, and for reference):

Reviewer's Responses to Questions

**Part I - Summary**

Reviewer #1: The authors have carefully and thoroughly addressed all issues I brought up in the previous review round. This is now a well-studied and supported research paper that should be of high interest to the field.

Reviewer #2: This is an interesting manuscript that investigates differences in CD4 T cell count at the time of HIV infection diagnosis or seroconversion, as a measure of virulence, in men who have sex with men (MSM) and heterosexuals (HET). The authors hypothesize that such differences are the result of differential selection bias during transmission and examine early CD4 T cell count measurements reported from 340,000 infected HSX individuals and MSM, across geographies, ethnicities and calendar year. The authors’ conclusion is that, because MSM have higher CD4 counts at the time of diagnosis and seroconversion than HET individuals, this reflects the greater selection bias for more fit viruses during penile/vaginal sex compared to anal sex. This increased “virulence” was not reflected in higher VL in HET individuals vs MSM, a fact not addressed in the study. Nevertheless, this large meta-analysis is novel and yields results that support differences in viral pathogenicity developing over time in different risk populations. The revisions made address all of this reviewers concerns.

**Part II – Major Issues: Key Experiments Required for Acceptance**

Reviewer #1: no issues remain

Reviewer #2: None

**Part III – Minor Issues: Editorial and Data Presentation Modifications**

Reviewer #1: (No Response)

Reviewer #2: None

PLOS authors have the option to publish the peer review history of their article (what does this mean?). If published, this will include your full peer review and any attached files.

Reviewer #1: **Yes: **Thomas Leitner

Reviewer #2: No

---

## [Editor Report · Acceptance letter]

17 Feb 2022

Dear Dr. Dixit,

We are delighted to inform you that your manuscript, "Transmitted HIV-1 is more virulent in heterosexual individuals than men-who-have-sex-with-men," has been formally accepted for publication in PLOS Pathogens.

Best regards,

Kasturi Haldar

Editor-in-Chief

PLOS Pathogens

orcid.org/0000-0001-5065-158X

Michael Malim

Editor-in-Chief

PLOS Pathogens

orcid.org/0000-0002-7699-2064